# Suppression of Hypoxia-Inducible Factor 1α by Low-Molecular-Weight Heparin Mitigates Ventilation-Induced Diaphragm Dysfunction in a Murine Endotoxemia Model

**DOI:** 10.3390/ijms22041702

**Published:** 2021-02-08

**Authors:** Li-Fu Li, Chung-Chieh Yu, Hung-Yu Huang, Huang-Pin Wu, Chien-Ming Chu, Chih-Yu Huang, Ping-Chi Liu, Yung-Yang Liu

**Affiliations:** 1Department of Internal Medicine, Division of Pulmonary and Critical Care Medicine, Chang Gung Memorial Hospital, Keelung 20401, Taiwan; lfp3434@cgmh.org.tw (L.-F.L.); ycc@cgmh.org.tw (C.-C.Y.); b9202071@cgmh.org.tw (H.-Y.H.); whanpyng@cgmh.org.tw (H.-P.W.); rocephen@cgmh.org.tw (C.-M.C.); hcu121@cgmh.org.tw (C.-Y.H.); ewind14@cgmh.org.tw (P.-C.L.); 2Department of Internal Medicine, Chang Gung University, Taoyuan 33302, Taiwan; 3Department of Respiratory Therapy, Chang Gung Memorial Hospital, Keelung 20401, Taiwan; 4Chest Department, Taipei Veterans General Hospital, Taipei 11217, Taiwan; 5Faculty of Medicine, School of Medicine, National Yang-Ming University, Taipei 11217, Taiwan; 6Institute of Clinical Medicine, School of Medicine, National Yang-Ming University, Taipei 11217, Taiwan

**Keywords:** endotoxemia, hypoxia-inducible factor-1α, low-molecular-weight heparin, mitochondria, ventilator-induced diaphragm dysfunction

## Abstract

Mechanical ventilation (MV) is required to maintain life for patients with sepsis-related acute lung injury but can cause diaphragmatic myotrauma with muscle damage and weakness, known as ventilator-induced diaphragm dysfunction (VIDD). Hypoxia-inducible factor 1α (HIF-1α) plays a crucial role in inducing inflammation and apoptosis. Low-molecular-weight heparin (LMWH) was proven to have anti-inflammatory properties. However, HIF-1α and LMWH affect sepsis-related diaphragm injury has not been investigated. We hypothesized that LMWH would reduce endotoxin-augmented VIDD through HIF-1α. C57BL/6 mice, either wild-type or HIF-1α–deficient, were exposed to MV with or without endotoxemia for 8 h. Enoxaparin (4 mg/kg) was administered subcutaneously 30 min before MV. MV with endotoxemia aggravated VIDD, as demonstrated by increased interleukin-6 and macrophage inflammatory protein-2 levels, oxidative loads, and the expression of HIF-1α, calpain, caspase-3, atrogin-1, muscle ring finger-1, and microtubule-associated protein light chain 3-II. Disorganized myofibrils, disrupted mitochondria, increased numbers of autophagic and apoptotic mediators, substantial apoptosis of diaphragm muscle fibers, and decreased diaphragm function were also observed (*p <* 0.05). Endotoxin-exacerbated VIDD and myonuclear apoptosis were attenuated by pharmacologic inhibition by LMWH and in HIF-1α–deficient mice (*p <* 0.05). Our data indicate that enoxaparin reduces endotoxin-augmented MV-induced diaphragmatic injury, partially through HIF-1α pathway inhibition.

## 1. Introduction

During sepsis, bacterial surface molecules, such as lipopolysaccharides (LPS), may evoke an inflammatory catastrophe with outbursts of inflammatory cytokines, including interleukin-6 (IL-6), macrophage inflammatory protein-2 (MIP-2), and tumor necrosis factor alpha (TNF-α), as well as vascular endothelial growth factor (VEGF), which can cause epithelial and endothelial damage, vascular leakage, lung edema, and hypoxemia, eventually resulting in multiple organ system failure [1,2,3,4,5,6]. Mechanical ventilation (MV) is indispensable for patients with acute respiratory failure, but it may also entail patient dependence on ventilators due to rapid deterioration of diaphragm muscle endurance and strength, known as ventilator-induced diaphragmatic dysfunction (VIDD) [7,8,9]. Accumulating evidence suggests that sepsis is a predominant cause of diaphragm weakness of patients in intensive care units (ICUs) [8,9,10,11]. Endotoxin is known to stimulate calpain activation influenced by the increased levels of oxidative stress in the diaphragm [12]. Sepsis-exacerbated diaphragm damage and VIDD are considered to share common molecular mechanisms, including increased oxidative stress, muscle proteolysis (emerging from calpain, caspase-3, autophagy–lysosomal pathway, and ubiquitin–proteasome system activation), and mitochondrial abnormalities within the diaphragm myofibrils, implying that sepsis may be a synergistic contributor to VIDD [12,13]. However, the mechanisms orchestrating the interactions among sepsis, MV, and these inflammatory cascades remain unclear.

Hypoxia-inducible factor 1α (HIF-1α) is a transcription factor consisting of an oxygen-regulated degradation domain and is ubiquitously expressed in all mammalian cell types [14]. The expression of HIF-1α is determined by the interplay between degradation and synthesis. Under hypoxic conditions, HIF-1α is synthesized and then conjugates with HIF-1β in the nucleus, where it is responsible for the transcription of various hypoxia-response genes, including erythropoietin and VEGF [14]. The expression of HIF-1α can also be stimulated under normoxic conditions by numerous factors, including exposure to LPS, inflammatory cytokines (IL-6, TNF-α, and VEGF), reactive oxygen species (ROS), and cyclic mechanical stretch in vitro or MV in vivo [1,3,11,15,16,17,18]. Under normoxic conditions, HIF-1α is hydroxylated by prolyl hydroxylases (PHDs) and detected by the von Hippel–Lindau ubiquitin ligase, which leads to polyubiquitination and proteasomal degradation [19].

Mounting studies have provided evidence that MV increases the expression of HIF-1α in animal models of diaphragm tissue [17,20,21]. During sepsis, LPS-induced HIF-1α activation promotes the production of inflammatory cytokines (including TNF-α, IL-1β, and IL-6) and proapoptotic mediators, resulting in inflammation and apoptosis [2,4,20]. Tissue hypoxia is frequently accompanied by inflammation in patients with sepsis, which is caused by several mechanisms, including hemodynamic compromise, regional arteriolar vasoconstriction, microvascular hypoperfusion, microthrombi formation related to hypercoagulability, and leucosequestration [2,4]. Mitochondria are a dominant source of diaphragmatic ROS in response to hypoxia and also upstream regulators controlling the HIF-1α signaling pathways facilitating diaphragm muscle injury during MV or endotoxemia [22,23].

Heparin, which has advantages in the treatment of sepsis owing to its anti-inflammatory and anticoagulant effects [24,25], can be classified into unfractionated or low-molecular-weight heparin (LMWH) [26]. LMWH is a promising heparin derivative because of its superior antithrombotic effects, better bioavailability and efficacy, and reduced risk of bleeding [26,27]. Enoxaparin, one type of LMWH, is safer and easier for clinical application than unfractionated heparin. In our prior study of an animal ventilator-induced lung injury (VILI) model, LMWH substantially mitigated microvascular permeability, inflammatory cell infiltration, the production of active plasminogen activator inhibitor-1 (PAI-1), and acute lung injury (ALI) scores owing to its anti-inflammatory effects and beneficial mechanisms neutralizing dysregulated coagulation and fibrinolysis [27]. However, the molecular mechanisms of action underlying the protection of LMWH in patients with sepsis-induced diaphragm injury have not been clarified.

Using a VIDD model of mice after LPS treatment, we investigated the effects of LMWH on (1) HIF-1α expression associated with the development of diaphragm damage during MV; (2) oxidative stress, inflammatory cytokine production, and endotoxin-enhanced diaphragm injury; (3) HIF signaling in cases of VIDD with sepsis; and (4) HIF signaling in cases of apoptosis of diaphragm muscle fibers with sepsis. We hypothesized that MV with or without LPS treatment would augment diaphragm dysfunction, the production of free radicals, and myonuclear apoptosis and that LMWH would ameliorate VIDD in endotoxemic mice through the HIF-1α pathway.

## 2. Results

### 2.1. Reduction of Endotoxin-Enhanced MV-Induced VIDD, Diaphragmatic Oxygen Radicals, and Inflammatory Cytokines by Enoxaparin

Mice were administered MV of either tidal volume (V_T_) = 6 mL/kg or 10 mL/kg with room air for 8 h to induce VIDD. The physiological conditions at the beginning and end of MV are presented in Appendix A. Stable hemodynamic status was maintained through monitoring of the mean arterial pressure of the mice. Transmission electron microscopy (TEM) was performed to explore MV- and LPS-induced changes in diaphragm ultrastructures. Our mitochondrial injury score, based on the morphological characteristics of mitochondria, represents the stages of cellular injury [28]. Compared with the other MV treatment groups and the non-ventilated controls, mice with endotoxemia subjected to V_T_ = 10 mL/kg exhibited increased disruptions in diaphragmatic myofibrillar structures, with larger lipid droplets, unclear A- and I-bands, damaged Z-bands, and mitochondrial swelling (Figure 1A–E). Notably, compared with V_T_ = 6 mL/kg groups and the non-ventilated controls, mice with V_T_ = 10 mL/kg exhibited significant diaphragmatic damage in endotoxemia (Figure 1A–E), which is in accordance with clinical low V_T_ protective ventilation strategy. The administration of enoxaparin substantially reduced damage to the diaphragmatic fibers (Figure 1F,G). Ultrasonography has emerged as a noninvasive tool for assessing the diaphragm in patients receiving MV [29,30]. To determine the effects of sepsis and MV on diaphragm contractile conditions, we measured diaphragm dysfunction by using small-animal ultrasound (VEVO 2100, Visual Sonics, Toronto, Canada). Decreased diaphragm excursion and thickening fraction were observed in mice with endotoxemia subjected to V_T_ = 10 mL/kg compared with the other MV treatment groups and the non-ventilated control mice (Figure 1H,I). The administration of enoxaparin substantially suppressed MV- and endotoxin-mediated increases in diaphragmatic weakness, suggesting it reduced muscle weakness induced by these two factors together. Several studies have demonstrated the critical role of MV-induced imbalance in terms of oxidative load, antioxidant capacity, and expression of inflammatory cytokines in inducing VIDD [1,12,29]. Increased levels of malondialdehyde (MDA), MIP-2, and IL-6 but decreased production of total antioxidant capacity was observed in mice with endotoxemia subjected to V_T_ = 10 mL/kg compared with the other MV treatment groups and the non-ventilated control mice (Figure 2A–D). However, a prevention of these features occurred after the administration of enoxaparin.

### 2.2. Suppression of Endotoxin-Augmented MV-Induced Diaphragmatic Calpain, Atrogin-1, and Murf-1 Expression and Autophagy by Enoxaparin

Western blot analyses were performed to identify the effects of MV on endotoxin-induced oxidative loads (calpain) and the ubiquitin–proteasome (atrogin-1 and muscle ring finger-1 [MuRF-1]) and autophagy–lysosomal (light chain 3-II [LC3-II] and Beclin-1) systems associated with VIDD. Total calpain, atrogin-1, MuRF-1, LC3-II, and Beclin-1 levels were higher in mice with endotoxemia subjected to V_T_ = 10 mL/kg than in mice in the other MV treatment groups and the non-ventilated control mice (Figure 2E and Figure 3). Enoxaparin administration substantially alleviated the enhanced expression of calpain, atrogin-1, MuRF-1, LC3-II, and Beclin-1 caused by endotoxemia and MV at V_T_ = 10 mL/kg.

### 2.3. Inhibition of Endotoxin-Stimulated MV-Induced Diaphragmatic HIF-1α mRNA and HIF-1α Protein Expression by Enoxaparin

HIF-1α is a transcription factor involved in the regulation of inflammatory cytokines in sepsis [14]. Real-time polymerase chain reaction (PCR) was performed to measure the effects of MV on endotoxin-associated HIF-1α mRNA expression in the diaphragm (Figure 4A). The upregulation of HIF-1α mRNA expression was more substantial in mice with endotoxemia subjected to V_T_ = 10 mL/kg than in mice in the other MV treatment groups and the control group. However, a prevention of this finding was observed in mice with endotoxemia and MV (V_T_ = 10 mL/kg) after the administration of enoxaparin (Figure 4A). Because HIF-1α upregulation was demonstrated to modulate stretch-induced ALI associated with multiorgan system failure, we measured HIF-1α expression to investigate the role of the HIF-1α pathway in VIDD (Figure 4B–D) [31]. Western blot analyses revealed increased HIF-1α expression in mice subjected to V_T_ = 10 mL/kg compared with mice in the other MV treatment groups and the control group. Moreover, the increase in HIF-1α expression in mice with endotoxemia treated with V_T_ = 10 mL/kg was substantially reduced by inhibition with enoxaparin (Figure 4B). We used immunohistochemistry to further confirm the amount of HIF-1α protein in endotoxin-aggravated VIDD (Figure 4C,D). Positive results of immunohistochemical staining for HIF-1α were substantially increased in the diaphragm muscle fibers of the mice with endotoxemia treated with V_T_ = 10 mL/kg compared with mice in the other MV treatment groups and the control group (Figure 4C,D). Consistent with the results of the western blot analyses, the increases in HIF-1α expression after MV were substantially mitigated by inhibition with enoxaparin (Figure 4C,D).

### 2.4. Reduction of Endotoxin-Enhanced VIDD in HIF-1α–Deficient Mice

HIF-1α–deficient mice were examined to determine the role of HIF-1α in stretch-induced diaphragm damage through examination of whether the exacerbation of diaphragm injuries following the administration of endotoxin was induced by HIF-1α expression. The effects of MV on different parameters—including an increase in oxidative stress and decreases in antioxidant levels, inflammatory cytokine generation, mitochondrial injury, autophagosome accumulation, diaphragm function, and expressions of calpain, atrogin-1, MuRF-1, and LC3-II—in mice with endotoxemia treated with V_T_ = 10 mL/kg were substantially weaker in HIF-1α–deficient mice (*p* < 0.05; Figure 5 and Figure 6). Furthermore, increases in diaphragmatic inflammation, oxidative stress, and mitochondrial damage were observed in the mice subjected to V_T_ = 10 mL/kg with endotoxin compared with those subjected to V_T_ = 10 mL/kg with normal saline and the non-ventilated control mice (Figure 5 and Figure 6), suggesting the combinatorial effects of LPS and MV.

### 2.5. Suppression of Endotoxin-Augmented MV-Induced Diaphragmatic Expression of Caspase-3 and BNIP-3 and Epithelial Apoptosis by Enoxaparin in HIF-1α-Deficient Mice

In addition to its role in oxidative stress, caspase-3 is vital for the intrinsic apoptotic pathway [28,32]. BCL2/adenovirus E1B 19 kDa protein–interacting protein 3 (BNIP-3) expression is transcriptionally upregulated by HIF-1α [33]. LPS-induced HIF-1α has been demonstrated to upregulate BNIP-3 and lead to increased generation of ROS, mitophagy, and programmed cell death. Capase-3 and BNIP-3 expression level measurements and terminal deoxynucleotidyl transferase-mediated dUTP-biotin nick end-labeling (TUNEL) staining were performed to explore the functions of the caspase-3 and BNIP-3 pathways and the apoptosis of diaphragm muscle fibers in endotoxin-aggravated VIDD (Figure 7). A substantial increase in caspase-3 and BNIP-3 expression levels and the appearance of TUNEL-positive apoptotic nuclei in diaphragm muscle fibers were observed in mice with endotoxemia treated with V_T_ = 10 mL/kg compared with mice in the other MV treatment groups and the control group (Figure 7). Notably, there was a reduction in MV and endotoxin-enhanced caspase-3 and BNIP-3 activities and apoptosis in diaphragm muscle fibers following the administration of enoxaparin. Levels of caspase-3 and BNIP-3 activity and apoptosis were also lower in the HIF-1α-deficient mice. However, the levels of apoptotic mediators in injured mice after enoxaparin or in HIF-1α-deficient mice were still higher than in control mice, suggesting other mechanistic pathways involved in apoptosis warrant further investigation. Our results suggest that inhibiting the HIF-1α pathway leads to an inhibition of MV- and endotoxin-induced oxidative and inflammatory processes in the diaphragm (Figure 8).

## 3. Discussion

Sepsis is a primary cause of mortality in ICU patients, and MV can provide adequate ventilation and oxygenation for patients with sepsis [8,9]. However, MV has been proven to evoke the production of proinflammatory cytokines and thus contribute to VIDD, and infection is a primary risk factor for the development of severe diaphragm weakness in critically ill patients receiving MV [29]. The prevalence of diaphragmatic dysfunction upon ICU admission was reported to be as high as 64%, implying that diaphragmatic dysfunction may account for unrecognized organ failure in patients with sepsis [34]. VIDD contributes to difficulty weaning from MV and increased ventilator-related mortality in patients with sepsis; however, effective therapeutic agents have yet to be identified. Previous rodent studies of VIDD have demonstrated that MV induces diaphragmatic injury through the generation of excessive ROS by activating protein oxidation, lipid peroxidation, and LC3, which causes subsequent diaphragm inactivity [32,33,35]. Moreover, sepsis- and MV-elicited ROS may promote the generation of inflammatory cytokines, including IL-6, MIP-2 in rodents, and VEGF [3,15,17,21]. These inflammatory mediators, which disturb diaphragmatic function during sepsis, could largely originate in the diaphragm itself, with subsequent autocrine/paracrine effects on skeletal muscle fibers [3,15,17,21]. In the present study, we applied our reported [28] animal model to investigate the effects and molecular pathways of LMWH to ameliorate VIDD in endotoxemic mice. We observed that LMWH can (1) reduce oxidative stress and restore antioxidant activity; (2) reduce the production of inflammatory cytokines MIP-2 and IL-6; (3) attenuate muscle proteolysis and apoptosis; (4) ameliorate mitochondrial injury and autophagy; and (5) restore ultrastructural integrity and diaphragm excursion and thickness in a mouse model of VIDD with endotoxemia. Furthermore, we studied the deleterious role of HIF-1α intervening in the pathogenic pathways of diaphragm injury.

HIF-1α is a pivotal regulator that adjusts cellular metabolism in response to hypoxic changes and controls the propagation and progression of inflammation [14]. Cyclic stretch by MV was reported to suppress succinate dehydrogenase (SDH) activity and increase succinate production, leading to the inhibition of PHDs and the reduction of polyubiquitination and proteasomal degradation and facilitating HIF-1α stabilization in vitro and in vivo [31]. In addition, prolonged MV evokes the upregulation of angiogenic/neogenetic mediators in accordance with HIF-1α gene expression in the diaphragm muscle [17]. The expression of HIF-1α can be activated interactively by multiple factors during MV, such as diaphragm disuse, inflammation, reduced diaphragm blood flow, and impaired oxygen delivery and uptake. Thus, ventilator-induced decline in diaphragmatic oxygenation could facilitate hypoxia-induced accumulation of ROS in the diaphragm muscle and lead to ventilator-induced diaphragm atrophy and impaired contractility [17,36]. In addition, a nonhypoxic mechanism was shown to mediate ROS-sensitive regulation of HIF-1α translocation and activation through the upregulation of inflammatory cytokines [37]. In this study, we demonstrated that MV activates HIF-1α gene expression, causing increased oxidative stress, inflammation, muscle proteolysis, and apoptosis in the diaphragms of endotoxemic mice. We also demonstrated that these detrimental effects were ameliorated in HIF-1α knockout mice. Notably, the results of the present study and of our recent publication [38] indicate that the activation of HIF-1α is not only responsive to intracellular oxygen concentration but is also elicited by oxidants, inflammatory cytokines, LPS, and cyclic stretch.

The acute HIF-1α activation stimulated by LPS has been demonstrated in both murine macrophages and human monocytes [2,39]. The activation of HIF-1α expression evoked by LPS under normoxic conditions may be caused by enhanced HIF-1α transcription and reduced HIF-1α degradation through PHD2 and PHD3 [2]. A modest reduction of arterial hypoxemia was noted in our animal model of endotoxemia with MV; however, HIF-1α expression was intensely activated in lung and diaphragm tissues under MV [38]. This may be explained by sepsis-related inflammatory hypoxia and mechanical stretch-induced normoxic stabilization of HIF-1α in alveolar epithelia, which are maintained by the inhibition of SDH [20,31]. Systemic inflammation in patients with sepsis receiving MV is associated with sepsis-induced tissue hypoxia and MV-induced decreased diaphragm blood flow and impeded oxygen delivery and uptake [36,40]. Furthermore, inflammatory cytokine production induced by LPS or MV can prohibit oxygen use by mitochondria and produce an imbalance between oxygen supply and demand, and ultimately the inflammatory tissues become extensively hypoxic, known as inflammation-associated hypoxia [20,36]. During sepsis, LPS-induced HIF-1α expression upregulates the production of ROS, inflammatory cytokines, and proapoptotic proteins (e.g., caspase-3, BNIP-3, and Beclin-1), contributing to inflammation and apoptosis [2,20]. Moreover, HIF-1α deletion in murine macrophages was shown to offer protective effects against LPS-induced mortality, and the blockade of HIF-1α activity has been proposed to be a therapeutic target for treating LPS-induced sepsis [2,20]. In the current study, HIF-1α deletion through knockout of the target gene was demonstrated to suppress MV- and LPS-induced oxidants, MIP-2, IL-6, and apoptotic mediators and restore antioxidant activity.

LMWH has been proven to prohibit LPS-induced ALI and repress systemic inflammation in animal models of sepsis [41,42]. Mounting evidence has highlighted the clinical importance of the cross-interaction between coagulation and inflammation in the pathogenesis of sepsis [43]. Heparin has been demonstrated to exert beneficial anti-inflammatory and anticoagulant effects in animal models of VILI and sepsis [27,38]. A meta-analysis of nine randomized clinical trials that included 465 patients indicated that adjuvant treatment with LMWH improves the PaO2/FiO2 index and improves the 28-day mortality rate in patients with ALI and acute respiratory distress syndrome (ARDS) [44]. Another clinical meta-analysis exploring the therapeutic effects of LMWH in patients with sepsis demonstrated that LMWH substantially restored patients’ platelet count and reduced their prothrombin time, Acute Physiologic Assessment and Chronic Health Evaluation II score, and 28-day mortality rate compared with standard therapy [26]. Furthermore, a recent randomized clinical trial revealed that therapeutic enoxaparin had the advantageous effects of improved gas exchange, reduced D-dimer levels, and a higher percentage of successful weaning from MV after respiratory failure in patients with severe COVID-19 [45]. Nevertheless, the actual molecular mechanisms of LMWH treatment for these critical illnesses have not yet been identified. LMWH was proven to mitigate pulmonary inflammation and fibrosis through the repression of HIF-1α and VEGF expression in an animal model of peritoneal fibrosis, supporting the hypothesis that LMWH offers benefits against hypoxia and inflammation [46]. Moreover, oxidative stress is provoked by mediators released from infiltrating inflammatory cells and reacts with redox-sensitive nuclear factor-κB, leading to the progression of inflammation and coagulation, which represents the pathogenesis of VIDD or sepsis [28,47]. In the present study, LMWH inhibited oxidative stress, inflammation, and apoptosis and increased antioxidant activity, eventually leading to recovery from pathological impairment through repressing HIF-1α. Furthermore, LMWH improved the contractile excursion and thickening fraction of damaged diaphragms; these parameters are commonly employed in the evaluation of diaphragm dysfunction in clinical practice [48,49].

In the present study, we demonstrated that MV with or without LPS induces diaphragm injury through autophagy and mitochondrial damage. Either LMWH or HIF-1α knockout can suppress MV-induced autophagy and mitochondrial dysfunction through the inhibition of autophagic biomarkers (LC3-II and Beclin-1) and mitochondrial injury of damaged diaphragm tissue. Various autophagy-related mediators, such as LC3-11 and Beclin-1, are upregulated in sepsis-induced ALI [50]. Beclin-1, a homologue to the mammalian yeast autophagy-related gene 6, is an essential autophagy-promoting gene required for the formation of autophagosomes and modulates cell survival through the process of autophagy [51,52,53]. During the initiation of autophagy, Beclin-1 incorporates with LC3-I, which converts to its membrane-bound form LC3-II, and collaborates with the ubiquitin-binding protein p62/sequestosome 1 [54]. The upregulation of BNIP-3 mRNA and protein levels is caused by a hypoxia-response element contained in the proximal promoter for BNIP-3 that combines with HIF-1, activating BNIP-3 expression [55]. Consequently, BNIP-3 overexpression may result in the progression of cell death through necrosis, autophagy, and apoptosis [55]. The activation of BNIP-3 drives it to move into the mitochondria, where it provokes the loss of mitochondrial membrane potential, produces ROS, and contributes to mitophagy. The pattern of mitochondrial dysfunction evoked by BNIP-3 is dependent upon the specific cell type. In cardiomyocytes, BNIP-3 translocates into the mitochondria, causing the release of cytochrome c and resulting in the activation of caspases and apoptosis [56].

In the current study, we use ultrasonography to perform morphological and functional assessments of diaphragm dysfunction induced by MV with or without LPS and provide evidence of the beneficial effects of LMWH on LPS-exacerbated VIDD in an animal model. Notably, diaphragmatic ultrasonography has been demonstrated to be a noninvasive, reliable, and convenient method for both morphological (e.g., the detection of muscle atrophy or thickness) and functional (e.g., excursion and motion) evaluation of diaphragm muscle, with proven high interobserver consistency [57,58]. Two ultrasonography parameters are widely used to assess diaphragmatic function: (1) diaphragmatic excursion, which represents the quality of diaphragm contractility [59] and (2) thickening fraction of the diaphragm, which reflects the quantity of muscle fibers in the diaphragm [60]. A reduction in diaphragmatic thickness was reported to be frequently associated with diaphragmatic weakness during MV [58]. This finding can be explained by prolonged MV evoking significant declines in both passive and active diaphragm contractile force by increasing proteolysis and reducing the generation of myofibrillar proteins [61]. Mounting studies describing the application of bedside ultrasonography assessment of the diaphragm in the process of discontinuing ventilator use suggest that both methods of measuring the thickening fraction of diaphragm muscles and excursion are reliable predictors of ventilator liberation and extubation outcomes [9,48,49].

Overall, we demonstrated that LMWH alleviates inflammation, oxidative stress, muscle proteolysis, apoptosis, autophagy, and mitochondrial injury and leads to significant amelioration of biochemical, pathological, and functional disturbances through the suppression of HIF-1α signaling. Understanding the relationships between HIF-1α and VIDD in a murine model of endotoxemia may be helpful in the development of potential therapeutic strategies targeting these pathways in the vital areas of ALI and ventilator dependence. Furthermore, identifying a specific LMWH that can mitigate VIDD in endotoxemic mice would provide insight into its application for the clinical treatment of VIDD in patients with sepsis receiving MV.

## 4. Materials and Methods

### 4.1. Experimental Animals

Wild-type or HIF-1α-deficient C57BL/6 mice, weighing between 20 and 25 g, aged between 6 and 8 weeks, were obtained from Jackson Laboratories (catalog number 007227, Bar Harbor, ME, USA) and National Laboratory Animal Center (Taipei, Taiwan) [62]. The HIF-1αfl/fl, mouse line was bred to C57BL/6 mice carrying a CD4cre transgene. In the resulting offspring, a region encompassing exon 2 was excised in CD^4+^ cells. Genotyping on tail DNA was performed as previously described [63]. An amplification of~250 bp by primers DP11 (5′-GCAGTTAAGAGCACTAGTTG) and DP12(5′-GGAGCTATCTCTCTAGACC) indicated the presence of a floxed HIF-1α allele. An amplification of ~200 bp indicated a wild-type HIF-1α allele. PCR was used to genotype tail DNA for the presence of CD4cre (forward 5′-CGATGCAACGAGTGATGAGG, reverse 5′-CGCATAACCAGTGAAACAGC).

CD4-Cre is the promoter element and HIF-1α is only knocked down in CD^4+^ cells [62]. The study was performed in strict accordance with the recommendations in the Guide for the Care and Use of Laboratory Animals of the National Institutes of Health (NIH). The protocol was approved by the Institutional Animal Care and Use Committee of Chang Gung Memorial Hospital (Permit number: 2017111001). All surgery was performed under xylazine and Zoletil anesthesia, and all efforts were made to minimize suffering.

### 4.2. Experimental Groups

Animals were randomly distributed into 7 groups in each experiment: group 1, non-ventilated control wild-type mice with normal saline; group 2, non-ventilated control wild-type mice with LPS; group 3, V_T_ 6 mL/kg wild-type mice with LPS; group 4, V_T_ 10 mL/kg wild-type mice with normal saline; group 5, V_T_ 10 mL/kg wild-type mice with LPS; group 6, V_T_ 10 mL/kg HIF-1α^−/−^ mice with LPS; group 7, V_T_ 10 mL/kg wild-type mice after enoxaparin (4 mg/kg) administration with LPS. In each group, three mice underwent TEM, and five mice underwent measurement for immunohistochemistry assay, inflammatory cytokines, TUNEL assay and western blots.

### 4.3. Lipopolysaccharide Administration

Mice will receive either 1 mg/kg of Salmonella typhosa lipopolysaccharide (Lot 81H4018; Sigma Chemical Co., St. Louis, MO, USA) or an equivalent volume of normal saline intravenously via the internal jugular vein as a control. After 1 h of spontaneous respiration to allow for development of a septic response, the mouse will be subjected to MV for 8 h [64,65].

### 4.4. Enoxaparin Administration

Enoxaparin (Sigma, St. Louis, MO, USA), 4 mg/kg, was given subcutaneously 30 min before MV, based on our previous in vivo study that revealed 4 mg/kg enoxaparin suppressed blood coagulation and lung injury without significant bleeding tendency [27,38].

### 4.5. Measurement of Diaphragm Excursion and Thickness

To determine the effects of MV and LPS on diaphragm contractile conditions, a small animal ultrasound (VEVO 2100, Visual Sonics, Toronto, Canada)) using a 40-MHz probe was used to measure B- and M-mode diaphragm excursion and thickness [29,30]. For diaphragm excursion, the probe was positioned below the right costal margin between the midclavicular and anterior axillary lines. For diaphragm thickening, the probe was positioned at the midaxillary line at the lower 1/3 area of chest. The arrows indicate the beginning and end of diaphragmatic contraction, and the distance between the arrows indicate diaphragm displacement (excursion). The diaphragm is visualized as a two-layered structure composed of two parallel echogenic layers of diaphragmatic pleura and peritoneal membranes sandwiching a nonechogenic layer of diaphragm muscle. Thickening fraction of diaphragm: thickness of end-inspiration—thickness of end-expiration/thickness of end-expiration.

### 4.6. Transmission Electron Microscopy

The diaphragms were fixed in 3% glutaraldehyde in 0.1 M cacodylate buffer (pH 7.4) for 1 h at 4 °C. The l diaphragms were then post-fixed in 1% osmium tetroxide (pH 7.4), dehydrated in a graded series of ethanol, and embedded in EPON-812. Thin sections (70 nm) were cut, stained with uranyl acetate and lead citrate, and examined on a Hitachi H-7500 EM transmission electron microscope (Hitachi, Ltd., Tokyo, Japan) 4.7. Mitochondrial Injury Score

Mitochondrial injury was semi-quantitatively measured by using a scoring system that is based on the characteristic ultrastructural characteristics of mitochondria attendant to the progressive stages of cellular injury [28]. The severity of ultrastructural damage was quantified by determining a composite score (based on the scale of 0–5) that represented all the mitochondria visualized within the microscopy field. Score 0 = normal appearance. Score 1 = swelling of endoplasmic reticulum, minimal mitochondrial swelling. Score 2 = mild mitochondrial swelling. Score 3 = moderate or focal high-amplitude swelling. Score 4 = diffuse high-amplitude swelling. Score 5 = high-amplitude swelling with mitochondrial flocculent densities or calcifications. An average number of 10 non-overlapping fields in TEM of diaphragm sections were analyzed for each section by a single investigator blinded to the mouse genotype.

### 4.7. Immunoblot Analysis

The diaphragms were homogenized in 0.5 mL of lysis buffer, as previously described [31]. Crude cell lysates were matched for protein concentration, resolved on a 10% bis-acrylamide gel, and electrotransferred to Immobilon-P membranes (Millipore Corp., Bedford, MA, USA). For the assay of atrogin-1, Beclin-1, BNIP-3, calpain, caspase-3, HIF-1α, LC3-II, MuRF-1, and GAPDH, Western blot analyses were performed with respective antibodies (New England BioLabs, Beverly, MA, USA and Santa Cruz Biotechnology, Santa Cruz, CA, USA,). Blots were developed by enhanced chemiluminescence (NEN Life Science Products, Boston, MA, USA).

### 4.8. Real-Time Polymerase Chain Reaction

For isolating total RNA, the diaphragms were homogenized in TRIzol reagents (Invitrogen Corporation, Carlsbad, CA, USA) according to the manufacturer’s instructions. Total RNA (1 μg) was reverse transcribed by using a GeneAmp PCR system 9600 (PerkinElmer, Life Sciences, Inc., Boston, MA, USA), as previously described [27]. The following primers were used for real-time polymerase chain reaction (PCR): HIF, forward primer 5′-GCAGCAGGAATTGGAACATT-3′ and reverse primer 5′-GCATGCTAAATCGGAGGGTA-3′; and GAPDH as internal control, forward primer 5′-AATGCATCCTGCACCACCAA-3′ and reverse primer 5′-gtagccatattcattgtcata-3′ (Integrated DNA Technologies, Inc., Coralville, IA, USA) [62,63]. All quantity PCR reactions using SYBR Master Mix were performed on an ABI Prism 7000 sequence detector PCR system (Applied Biosystems, Foster City, CA, USA).

### 4.9. Analysis of Data

The Western blots were quantitated using an NIH image analyzer Image J 1.27z (National Institutes of Health, Bethesda, MD, USA) and presented as arbitrary units. Values were expressed as the mean ± SD from at least 5 separate experiments. The data of protein oxidation, histopathologic assay, and oxygenation were analyzed using Statview 5.0 (Abascus Concepts, Cary, NC, USA; SAS Institute). All results of real-time PCR and Western blots were normalized to the non-ventilated control wild-type mice with LPS. ANOVA was used to assess the statistical significance of the differences, followed by multiple comparisons with a Scheffe′s test, and a *p* value < 0.05 was considered statistically significant. Additional details, including measurement of cytokines in bronchoalveolar lavage fluid, immunoblot analysis, immunohistochemistry, TUNEL assay, and ventilator protocol were performed as previously described [27,28,38].

## Figures and Tables

**Figure 1 ijms-22-01702-f001:**
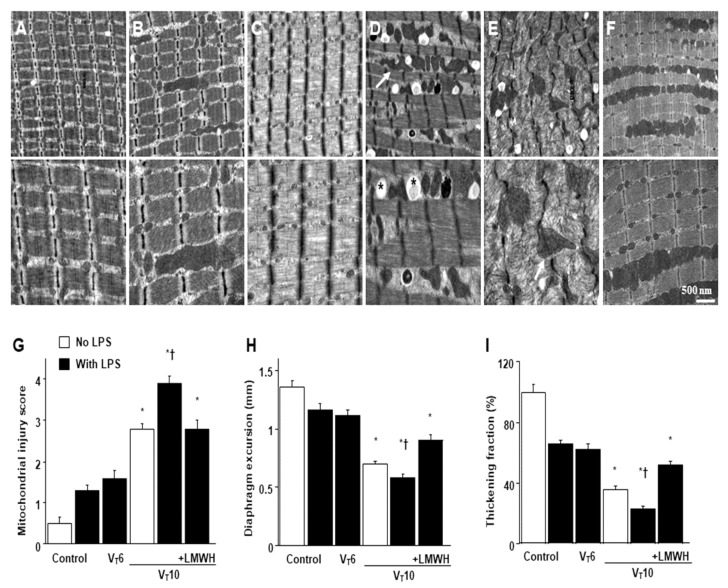
Electron microscopy, excursion, and thickening of the diaphragm. Representative micrographs of the longitudinal sections of diaphragm (×20,000: upper panel; ×40,000: lower panel) were from the same diaphragms of non-ventilated control mice and mice ventilated at a tidal volume (V_T_) of 6 mL/kg (V_T_ 6) or 10 mL/kg (V_T_ 10) for 8 h with or without LPS administration (n = 3 per group). (**A**,**B**) Non-ventilated control wild-type mice with or without LPS treatment: normal sarcomeres with distinct A bands, I bands, and Z bands; (**C**) 6 mL/kg wild-type mice with LPS treatment: reduction of diaphragmatic disruption compared to that of 10 mL/kg groups; (**D**) 10 mL/kg wild-type mice without LPS treatment (normal saline): increase of diaphragmatic disarray; (**E**) 10 mL/kg wild-type mice with LPS treatment: disruption of sarcomeric structure with loss of streaming of Z bands, mitochondrial swelling, and accumulation of lipid droplets (asterisks); (**F**) 10 mL/kg wild-type mice pretreated with enoxaparin: attenuation of diaphragmatic disruption. (**G**) Injury scores of mitochondria were from the diaphragms of non-ventilated control mice and mice ventilated at a tidal volume of 6 mL/kg or 10 mL/kg for 8 h with or without LPS administration (n = 3 per group). (**H**,**I**) Excursion and thickness variation of diaphragm. Mitochondrial swelling with concurrent loss of cristae and autophagosomes containing heterogeneous cargo are identified by arrows. Enoxaparin, 4 mg/kg, was given subcutaneously 30 min before mechanical ventilation. * *p* < 0.05 versus the non-ventilated control mice with LPS treatment; † *p* < 0.05 versus all other groups. Scale bar represents 500 nm. LPS = lipopolysaccharide; LMWH = low-molecular-weight heparin.

**Figure 2 ijms-22-01702-f002:**
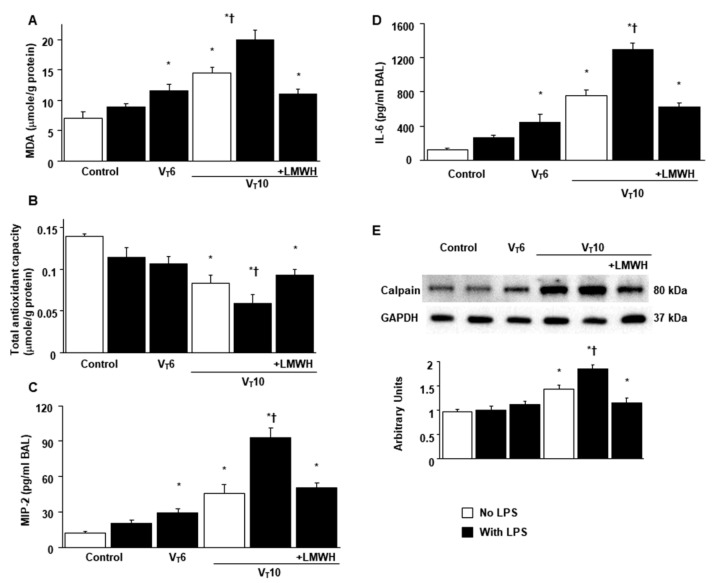
Inhibition of endotoxin-aggravated mechanical ventilation-enhanced oxidative stress, inflammatory cytokines production, and calpain expression by enoxaparin. (**A**) MDA (diaphragm), (**B**) total antioxidant capacity (diaphragm), (**C**) BAL fluid MIP-2, and (**D**) BAL fluid IL-6 were from the non-ventilated control mice and mice ventilated at a tidal volume of 6 mL/kg or 10 mL/kg for 8 h with or without LPS administration (n = 5 per group). Western blots were performed using antibodies that recognize calpain (**E**) and GAPDH expression from the diaphragms of non-ventilated control mice and mice ventilated at a tidal volume of 6 mL/kg or 10 mL/kg for 8 h with or without LPS administration (n = 5 per group). Arbitrary units were expressed as relative calpain activation (n = 5 per group). Enoxaparin, 4 mg/kg, was given subcutaneously 30 min before mechanical ventilation. * *p* < 0.05 versus the non-ventilated control mice with LPS treatment; † *p* < 0.05 versus all other groups. BAL = bronchoalveolar lavage; GAPDH = glyceraldehydes-phosphate dehydrogenase; IL = interleukin; MIP-2 = macrophage inflammatory protein-2; MDA = malondialdehyde.

**Figure 3 ijms-22-01702-f003:**
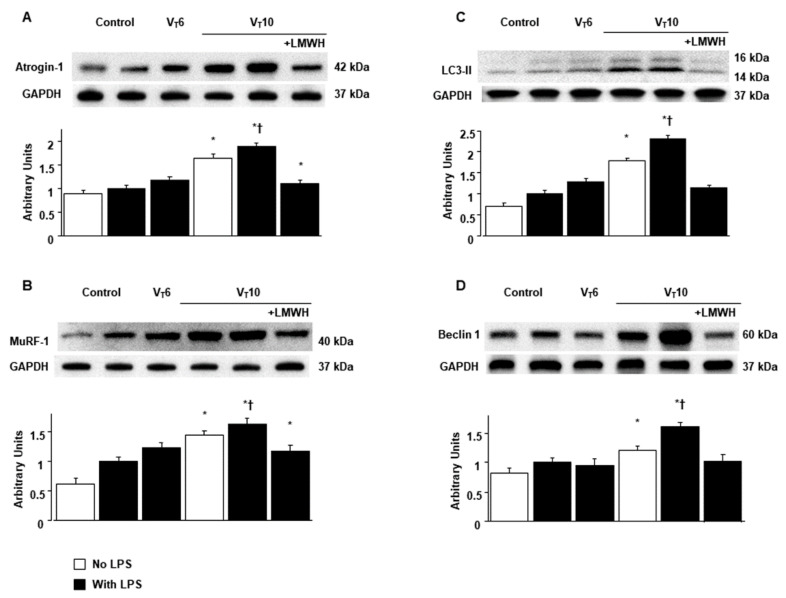
Reduction of endotoxin-augmented mechanical ventilation-mediated atrogin-1, MuRF-1, LC3-II, and Beclin expression by enoxaparin. Western blots were performed using antibodies that recognize atrogin-1 (**A**), MuRF-1 (**B**), LC3-II (**C**), beclin (**D**), and GAPDH expression from the diaphragms of non-ventilated control mice and mice ventilated at a tidal volume of 6 mL/kg or 10 mL/kg for 8 h with or without LPS administration (n = 5 per group). Arbitrary units were expressed as relative atrogin-1, MuRF-1, LC3-II, and beclin activation (n = 5 per group). Enoxaparin, 4 mg/kg, was given subcutaneously 30 min before mechanical ventilation. * *p* < 0.05 versus the non-ventilated control mice with LPS treatment; † *p* < 0.05 versus all other groups. LC3-II = light chain 3-II; MuRF-1 = muscle ring finger-1.

**Figure 4 ijms-22-01702-f004:**
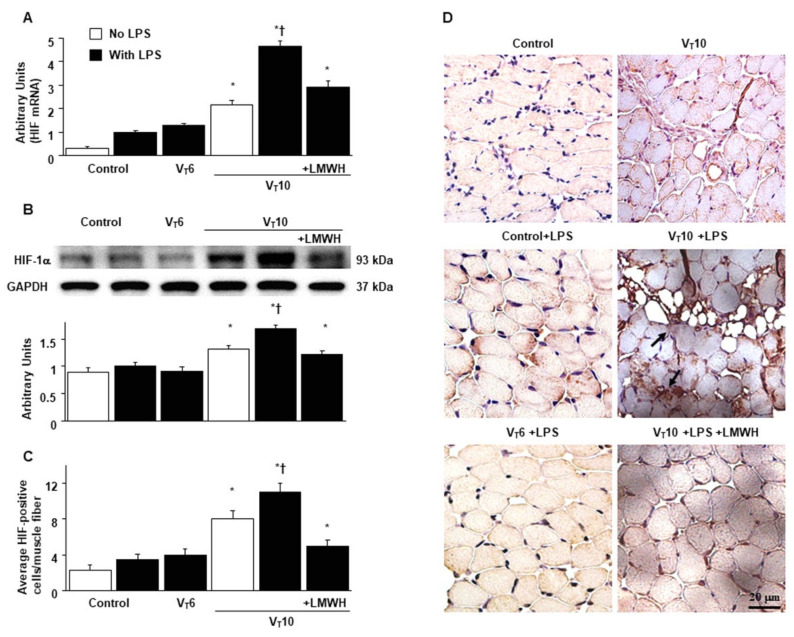
Suppression of endotoxin-augmented mechanical ventilation-induced HIF-1α mRNA activation and HIF-1α protein expression by enoxaparin. (**A**) Real-time PCR performed for HIF-1α mRNA expression was from the diaphragms of non-ventilated control mice and mice ventilated at a tidal volume of 6 mL/kg or 10 mL/kg for 8 h with or without LPS administration (n = 5 per group). Arbitrary units were expressed as the ratio of HIF-1α mRNA to GAPDH (n = 5 per group). (**B**) Western blots from the same animals were conducted using antibodies that recognize HIF and GAPDH expression from the diaphragms of non-ventilated control mice and mice ventilated at a tidal volume of 6 mL/kg or 10 mL/kg for 8 h with or without LPS administration (n = 5 per group). Arbitrary units were expressed as the ratio of HIF-1α to GAPDH (n = 5 per group). (**C** and **D**) Representative micrographs (×400) with HIF-1α staining of paraffin lung sections and quantification were from non-ventilated control mice and mice ventilated at a tidal volume of 6 mL/kg or 10 mL/kg for 8 h with or without LPS administration (n = 5 per group). Enoxaparin, 4 mg/kg, was given subcutaneously 30 min before mechanical ventilation. Scale bars represent 20 μm. * *p* < 0.05 versus the non-ventilated control mice with LPS treatment; † *p* < 0.05 versus all other groups.

**Figure 5 ijms-22-01702-f005:**
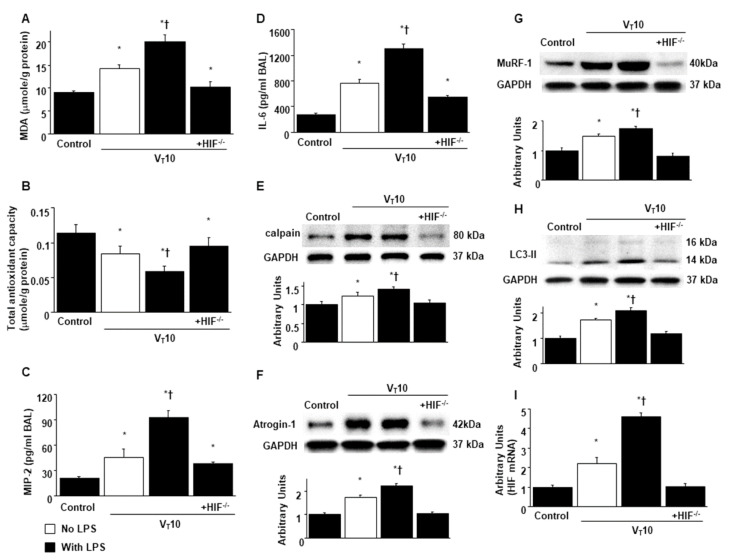
Abrogation of endotoxin-stimulated mechanical ventilation-mediated diaphragm dysfunction in HIF-1α deficient mice. (**A**) MDA (diaphragm), (**B**) total antioxidant capacity (diaphragm), (**C**) BAL fluid MIP-2, and (**D**) BAL fluid IL-6 were from the non-ventilated control mice and mice ventilated at a tidal volume of 10 mL/kg for 8 h with or without LPS administration (n = 5 per group). Western blots were performed using antibodies that recognize calpain (**E**), atrogin-1 (**F**), MuRF-1 (**G**), LC3-II (H), and GAPDH expression from the diaphragms of non-ventilated control mice and mice ventilated at a tidal volume of 10 mL/kg for 8 h with or without LPS administration (n = 5 per group). Arbitrary units were expressed as relative calpain, atrogin-1, MuRF-1, and LC3-II activation (n = 5 per group). (**I**) Real-time PCR performed for HIF-1α mRNA expression was from the diaphragms of non-ventilated control mice and mice ventilated at a tidal volume of 10 mL/kg for 8 h with or without LPS administration (n = 5 per group). Arbitrary units were expressed as the ratio of HIF-1α mRNA to GAPDH (n = 5 per group). * *p < 0.05* versus the non-ventilated control mice with LPS; † *p < 0.05* versus HIF-1α-deficient mice. HIF^−/−^ = hypoxia inducible factor-1α-deficient mice.

**Figure 6 ijms-22-01702-f006:**
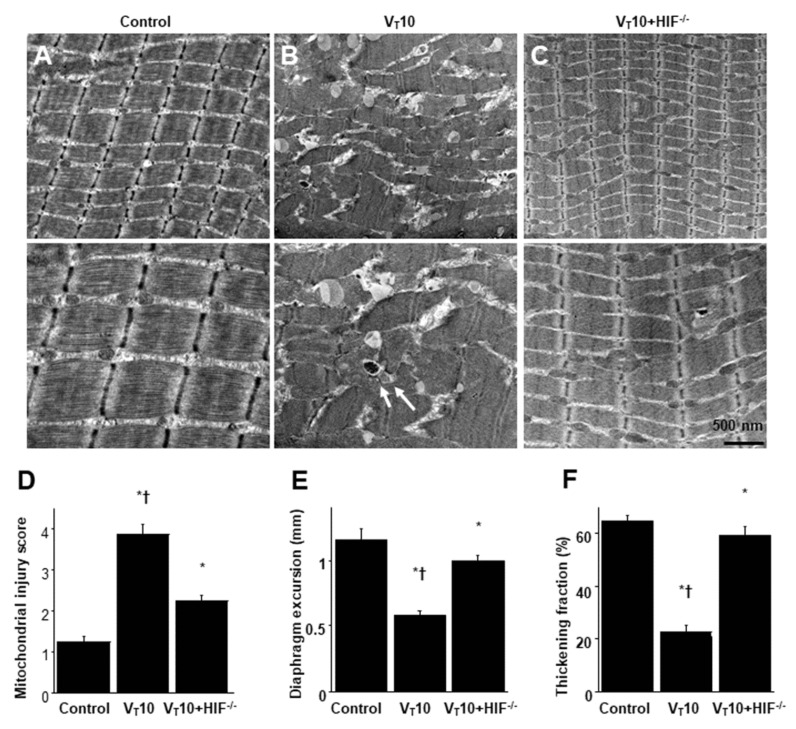
Reduction of endotoxin-exacerbated mechanical ventilation-induced diaphragm and mitochondrial injury in HIF-1α deficient mice. (**A**–**D**) Representative micrographs of the longitudinal sections of diaphragm (×20,000: upper panel; ×40,000: lower panel) were from the diaphragms of non-ventilated control mice and mice ventilated at a tidal volume of 10 mL/kg for 8 h with LPS administration (n = 3 per group). Mitochondrial swelling with coexisting vacuole formation, loss of cristae, and autophagosomes containing heterogeneous cargo are identified by arrows. Mitochondrial swelling with concurrent loss of cristae and autophagosomes containing heterogeneous cargo are identified by arrows. (**E**,**F**) Excursion and thickness variation of diaphragm. * *p* < 0.05 versus the non-ventilated control mice with LPS treatment; † *p < 0.05* versus HIF-1α-deficient mice. Scale bars represent 500 nm.

**Figure 7 ijms-22-01702-f007:**
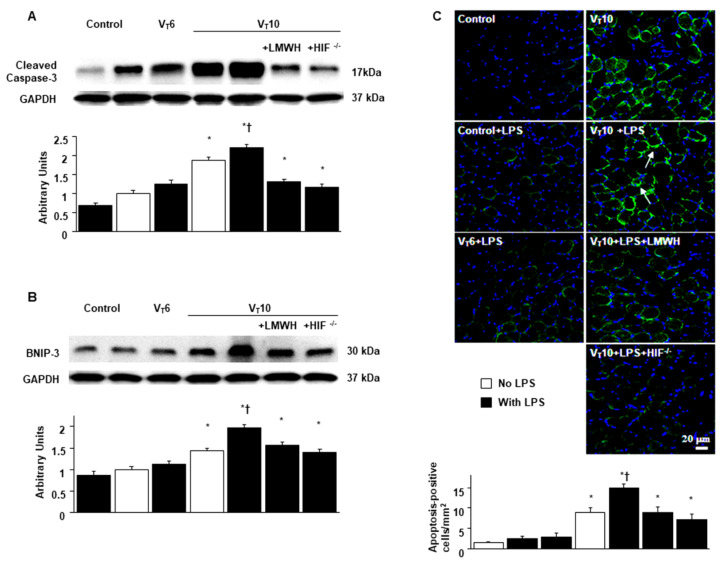
Suppression of endotoxin-augmented mechanical ventilation-induced expression of caspase-3 and BNIP-3, and muscle fiber apoptosis by enoxaparin and in HIF deficient mice. Caspase-3 (**A**), BNIP-3 (**B**), and GAPDH expression from the diaphragms of non-ventilated control mice and mice ventilated at a tidal volume of 6 mL/kg or 10 mL/kg for 8 h with or without LPS administration (n = 5 per group). Arbitrary units were expressed as relative cleaved caspase-3 and BNIP-3 activation (n = 5 per group). (**C**) Representative micrographs (×400) with TUNEL staining of paraffin diaphragm sections and quantification were from the diaphragms of non-ventilated control mice and mice ventilated at a tidal volume of 6 mL/kg or 10 mL/kg for 8 h with or without LPS administration (n = 5 per group). Enoxaparin, 4 mg/kg, was given subcutaneously 30 min before mechanical ventilation. Apoptotic cells are identified by arrows. A bright green signal indicates positive staining of apoptotic cells, and shades of dull green signify non-reactive cells. * *p* < 0.05 versus the non-ventilated control mice with room air; † *p* < 0.05 versus all other groups. Scale bars represent 20 μm. BNIP-3 = BCL2/adenovirus E1B 19 kDa protein-interacting protein 3; TUNEL = terminal deoxynucleotidyl transferase-mediated dUTP-biotin nick end-labeling.

**Figure 8 ijms-22-01702-f008:**
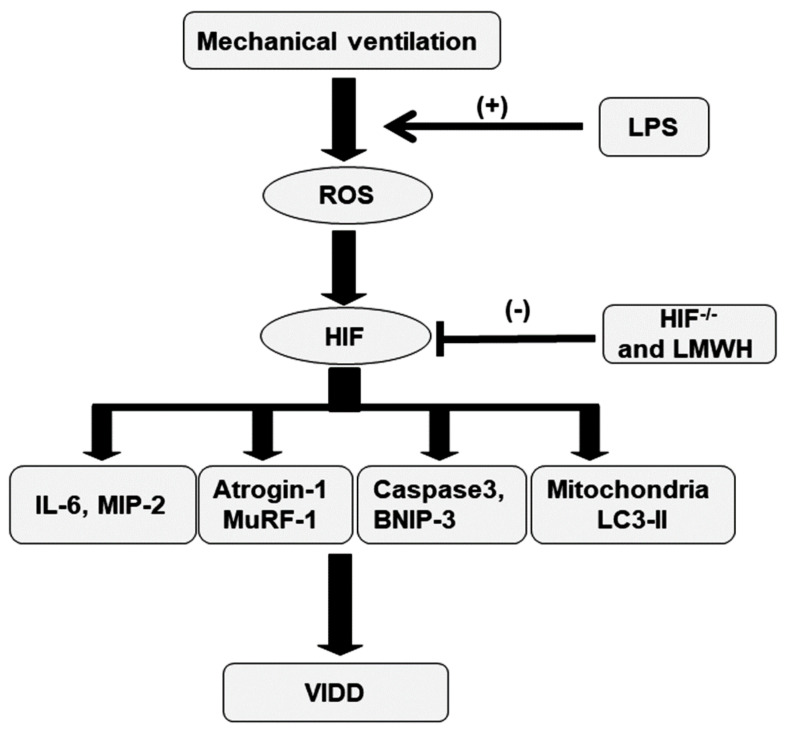
Schematic figure illustrating the signaling pathway activation with. mechanical ventilation and endotoxemia. Endotoxin-induced augmentation of mechanical stretch-mediated cytokine production and diaphragm damage were attenuated by the administration of enoxaparin and with HIF-1α homozygous knockout. HIF = hypoxia-inducible factor; IL = interleukin; LC3-II = light chain 3-II; LMWH = low-molecular-weight heparin; LPS = lipopolysaccharide; MIP-2 = macrophage inflammatory protein-2; MuRF-1 = muscle ring finger-1; ROS = reactive oxygen species; BNIP-3 = BCL2/adenovirus E1B 19 kDa protein-interacting protein 3; VIDD = ventilator-induced diaphragm dysfunction.

## Data Availability

The data presented in this study are available on request from the corresponding author.

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
