# Peer review of "Suppression of Hypoxia-Inducible Factor 1α by Low-Molecular-Weight Heparin Mitigates Ventilation-Induced Diaphragm Dysfunction in a Murine Endotoxemia Model"

_ijms, 2021, doi:10.3390/ijms22041702_

Round 1

Reviewer 1 Report

The authors investigated the effect of endotoxin and mechanical ventilation (MV) on the function, morphology and physiology of the diaphragm in mice. They could show that MV and endotoxin impair the function, morphology and physiology of the diaphragm. This effect could be prevented by enoxaparin (which decreased HIF-1α) and HIF-1α knock-out, showing the central role of HIF-1α in this condition.

I have the following questions and comments:

  1. Presentation of the paper: Figures and figure legends have to be included into the paper as described in the instructions to the authors.
  2. Methods: A description of the preparation of western blots and PCR is lacking. Also the antibodies used must be described. The application of enoxaparin to the mice should also be described. I assume that enoxaparin was administered immediately with the endotoxin or MV. This means that enoxaparin is preventive, not curative. This should be considered when the action of enoxaparin is described in the entire manuscript (e.g. enoxaparin does not reverse but prevent oxidative damage).
  3. The discussion is too long. I believe that it could easily be cut by 50%.
  4. Mitochondrial damage: was quantified using a scoring system, which quantifies the morphology, but not the function. This is ok, but should be stated in the result section. Since most ROS stem from mitochondria, this should be indicated in Fig. 8.
  5. Figure 7C: The positivity of the TUNEL assay is difficult to observe. There may be images, where this can be seen better.
  6. Discussion: the authors could at least speculate how enoxaparin could work. At least, they observed a decrease in HIF-1α in the presence of enoxaparin, which may explain its function.

Author Response

The authors investigated the effect of endotoxin and mechanical ventilation (MV) on the function, morphology and physiology of the diaphragm in mice. They could show that MV and endotoxin impair the function, morphology and physiology of the diaphragm. This effect could be prevented by enoxaparin (which decreased HIF-1α) and HIF-1α knock-out, showing the central role of HIF-1α in this condition.

I have the following questions and comments:

  •  Presentation of the paper: Figures and figure legends have to be included into the paper as described in the instructions to the authors.

Ans: we have done this according to the description in the instructions to the authors.

  • Methods: A description of the preparation of western blots and PCR is lacking. Also the antibodies used must be described. The application of enoxaparin to the mice should also be described. I assume that enoxaparin was administered immediately with the endotoxin or MV. This means that enoxaparin is preventive, not curative. This should be considered when the action of enoxaparin is described in the entire manuscript (e.g. enoxaparin does not reverse but prevent oxidative damage).

Ans: (1) we have added the description of the preparation of western blots and PCR and used antibodies into the Methods; (2) we have added the application of enoxaparin to the mice into the Methods New 4.4; (3) we have changed the use of “reversal” to “prevention” in line 124 and line 143.

  • The discussion is too long. I believe that it could easily be cut by 50%.

Ans: we have cut the redundant parts (line 199-204, line 220-223, line 239-240, line 259, line 267-271, line 318-320) in the revised manuscript.

  • Mitochondrial damage: was quantified using a scoring system, which quantifies the morphology, but not the function. This is ok, but should be stated in the result section. Since most ROS stem from mitochondria, this should be indicated in Fig. 8.

Ans: (1) we have added the description “Our mitochondrial injury score, based on the morphological characteristics of mitochondria, represents the stages of cellular injury [28].” in line 107; (2) we have revised the Figure 8 and describe as “Mitochondria are a dominant source of diaphragmatic ROS in response to hypoxia” in Introduction (paragraph 3, line 10-11).

  • Figure 7C: The positivity of the TUNEL assay is difficult to observe. There may be images, where this can be seen better.

Ans: we have revised the Figure 7 to improve the quality of TUNEL stain.

  • Discussion: the authors could at least speculate how enoxaparin could work. At least, they observed a decrease in HIF-1α in the presence of enoxaparin, which may explain its function.

Ans: we have explained this as “In the present study, LMWH inhibited oxidative stress, inflammation, and apoptosis and increased antioxidant activity, eventually leading to recovery from pathological impairment through repressing HIF-1a.” in Discussion (paragraph 4, line 28-31).

Reviewer 2 Report

The paper of Li-Fu Li et al. presents results of experiments suggesting that low-molecular-weight heparin reduces mechanical injury of diaphragm acting on a Hif1α-dependent pathway.

In general, the paper is quite interesting and results are worth publishing but there are some points which need attention/clarification.

First of all, the authors shoud clarify what was the reason for including the „VT 6 mL/kg wild-type mice with LPS” into the experiments? Since there was no control group „VT 6 mL/kg wild-type mice with saline”, thus, any observations made in „VT 6 mL/kg wild-type mice with LPS” group might reflect the effect of LPS or, just as well, mechanical ventillation.

What is even more confusing, in the legend to Fig. 1., the authors present sarcomeric structure in control (non-ventilated) mice +/- LPS in A and B and then write:

“C) mL/kg wild-type mice with LPS treatment: reduction of diaphragmatic disruption”. Does it mean that LPS together with VT 6 mL/kg in fact has some protective effects? If so, authors should explain a possible mechanism. However, I feel that none of the presented results support this “protective” effect.

Minor:

  1. Despite the existence of the Abbreviation lists, all abbreviations should be explained with their first usage in the main text (e.g. PAI-1, MDA).
  2. Line 89-90: the word „action” is missing from the sentence: ”However, the molecular mechanisms of LMWH in patients...”, as there is no such thing as „molecular mechanism of LMWH”.
  3. Lines 70-90: Although in general, the paper is well-written the paragraph is a bit chaotic and should be reorganized.
  4. Line 129: a word of clarification is needed (maybe in the Introduction, adding an adequate sentence and citation to information on molecular mechanisms of sepsis-exacerbated diaphragm damage; line 48 and on) why the authors considered a protease – calpain – as the oxidative load marker.
  5. Line 118: in the sentence “The administration of enoxaparin substantially suppressed MV- and endotoxin- mediated increases in diaphragmatic weakness.” The plural word “increases” might suggest that enoxaparin reduced both the MV-mediated and the endotoxin-mediated diaphragmatic injury, when in fact, it reduced weakness induced by the two factors together.
  6. In line 142 and on the authors state: “The upregulation of HIF-1a mRNA expression was more substantial in mice with endotoxemia subjected to VT = 10 mL/kg than in mice in the other MV treatment groups and the control group. However, a reversal of this finding was observed after the administration of enoxaparin (Figure 4A).”

In such a case, “a reversal” would mean that after enoxaparin treatment of VT=10 mL/kg + LPS mice, the upregulation of Hif1a was less substantial than in other groups when in fact, it was only lower than in untreated VT=10 mL/kg + LPS mice.

  1. Line 151: “Hif1a activation” suggests measurements of the protein transcriptional (or any other) activity, when actually the amount of the protein is assessed.
  2. Line 160: “…examination of whether the healing of diaphragm injuries following the administration of endotoxin was induced by HIF-1a expression”. I thought that the working hypothesis was that Hif1a expression exacerbates injury…
  3. Line 170: it should be: combinatorial effects of LPS and MV.
  4. Line 235 and on: “In this study, we demonstrated that MV, with or without LPS, activates HIF-1a gene expression, causing increased oxidative stress, inflammation, muscle proteolysis, and apoptosis in the diaphragms of endotoxemic mice.” The authors should decide: “with or without LPS” or “endotoxemic mice” (i.e., with LPS, but not without it).
  5. Line 354: check the formatting please.
  6. Table S1: it should be pH instead of PH.
  7. Figure 1: lipid droplets could be marked somehow.
  8. Figure 7: the level of cleaved caspase 3 and BNIP-3 in Hif1a-deficient mice +ventilation+LPS is still higher than in control mice. Similarly, the addition of LMWH only partially supress apoptotic mediators. It should be therefore clarified in the text (e.g., line 188 and 265).
  9. The description of some methods (TEM, Western blot) is not adequate. There are no details of procedures and used antibodies.

Round 2

Reviewer 1 Report

No further requests

Author Response

Thanks for your corrections.

Reviewer 2 Report

The revised manuscript can be accepted provided the authors correct the sentence „However, a prevention of this finding was observed in mice with endotoxemia and MV (VT = 10 mL/kg) after the administration of enoxaparin (Figure 4A).” (line 194 in the corrected version). I do not think that they planned to state that sb/sth prevented them from making a finding, it is probably a slip of the pen...

I would suggest something like: „However, administration of enoxaparin to mice with endotoxemia and MV (VT = 10 mL/kg) prevented this change.”

Please remove „4.7. Mitochondrial Injury Score” from the line 512.

Author Response

1.However, a prevention of this finding was observed in mice with endotoxemia and MV (VT = 10 mL/kg) after the administration of enoxaparin (Figure 4A).” (line 194 in the corrected version). I do not think that they planned to state that sb/sth prevented them from making a finding, it is probably a slip of the pen...

Ans: we have revised as the reviewer’s comment “However, administration of enoxaparin to mice with endotoxemia and MV (VT = 10 mL/kg) prevented this change (Figure 4A)” in the revised manuscript, line 194.

2. Please remove ,4.7. Mitochondrial Injury Score” from the line 512.

Ans. We have moved this title to its correct line.
